# Selective Extracellular Signal-Regulated Kinase 1/2 (ERK1/2) Inhibition by the SCH772984 Compound Attenuates In Vitro and In Vivo Inflammatory Responses and Prolongs Survival in Murine Sepsis Models

**DOI:** 10.3390/ijms221910204

**Published:** 2021-09-22

**Authors:** Michal Kopczynski, Izabela Rumienczyk, Maria Kulecka, Małgorzata Statkiewicz, Kazimiera Pysniak, Zuzanna Sandowska-Markiewicz, Urszula Wojcik-Trechcinska, Krzysztof Goryca, Karolina Pyziak, Eliza Majewska, Magdalena Masiejczyk, Katarzyna Wojcik-Jaszczynska, Tomasz Rzymski, Karol Bomsztyk, Jerzy Ostrowski, Michal Mikula

**Affiliations:** 1Department of Genetics, Maria Sklodowska-Curie National Research Institute of Oncology, 02-781 Warsaw, Poland; michal.kopczynski@pib-nio.pl (M.K.); izabela.rumienczyk@pib-nio.pl (I.R.); mkulecka@cmkp.edu.pl (M.K.); malgorzata.statkiewicz@pib-nio.pl (M.S.); pysniak27@interia.pl (K.P.); zuzanna.sandowska-markiewicz@pib-nio.pl (Z.S.-M.); ula.wojciktrech@gmail.com (U.W.-T.); jostrow@warman.com.pl (J.O.); 2Department of Gastroenterology, Hepatology and Clinical Oncology, Centre for Postgraduate Medical Education, 01-813 Warsaw, Poland; 3Genomics Core Facility, Centre of New Technologies, University of Warsaw, 02-097 Warsaw, Poland; kgoryca@gmail.com; 4Biology R&D, Ryvu Therapeutics S.A., 30-394 Krakow, Poland; karolina.pyziak@ryvu.com (K.P.); elizazmajewska@gmail.com (E.M.); magdalena.masiejczyk@ryvu.com (M.M.); katarzyna.wojcik@ryvu.com (K.W.-J.); tomasz.rzymski@ryvu.com (T.R.); 5UW Medicine South Lake Union, University of Washington, Seattle, WA 98109, USA; karolb@uw.edu

**Keywords:** sepsis, drugs repurposing, ERK1/2, SCH772984, cecal ligation and puncture

## Abstract

Sepsis is the leading cause of death in intensive care units worldwide. Current treatments of sepsis are largely supportive and clinical trials using specific pharmacotherapy for sepsis have failed to improve outcomes. Here, we used the lipopolysaccharide (LPS)-stimulated mouse RAW264.7 cell line and AlphaLisa assay for TNFa as a readout to perform a supervised drug repurposing screen for sepsis treatment with compounds targeting epigenetic enzymes, including kinases. We identified the SCH772984 compound, an extracellular signal-regulated kinase (ERK) 1/2 inhibitor, as an effective blocker of TNFa production in vitro. RNA-Seq of the SCH772984-treated RAW264.7 cells at 1, 4, and 24 h time points of LPS challenge followed by functional annotation of differentially expressed genes highlighted the suppression of cellular pathways related to the immune system. SCH772984 treatment improved survival in the LPS-induced lethal endotoxemia and cecal ligation and puncture (CLP) mouse models of sepsis, and reduced plasma levels of Ccl2/Mcp1. Functional analyses of RNA-seq datasets for kidney, lung, liver, and heart tissues from SCH772984-treated animals collected at 6 h and 12 h post-CLP revealed a significant downregulation of pathways related to the immune response and platelets activation but upregulation of the extracellular matrix organization and retinoic acid signaling pathways. Thus, this study defined transcriptome signatures of SCH772984 action in vitro and in vivo, an agent that has the potential to improve sepsis outcome.

## 1. Introduction

Sepsis, a systemic host’s response to pathogens, is a multifaceted infection-induced syndrome of physiologic, pathologic, and biochemical abnormalities [1]. More than 49 million sepsis cases and 11 million sepsis-related deaths are estimated to occur annually worldwide [2]. The mortality from sepsis remains extremely high and is the leading cause of death in intensive care units (ICU) [3]. Furthermore, sepsis survivors often experience long-term physical, psychological, or cognitive disabilities [4]. Therefore, sepsis confers a substantial health and economic burden to health care systems. Homeostasis of the pro- and anti-inflammatory mechanisms is essential for maintaining the balance between protective and tissue-damaging inflammatory responses [1]. If the inflammatory reaction becomes dysregulated, activation of the immune system becomes enormous and results in systemic inflammatory response syndrome (SIRS) that could ultimately lead to life-threatening severe sepsis associated with organ dysfunction [4]. Current treatments of sepsis are largely supportive and consist of the application of broad-spectrum antibiotics, and optimization of the intravascular volume to maintain blood pressure and to support organ function [1,5]. Sepsis treatment has remained essentially unchanged in decades [3,6]. Therefore, there is a need for new therapeutic approaches to treat this devastating syndrome.

In response to pathogen infection, the activation of pattern recognition receptors (PRRs) [7], including the Toll-like receptors (TLRs), on the host’s cells surface activates multiple downstream signaling pathways, including the members of each of the major mitogen-activated protein kinase (MAPK) subfamilies—the extracellular signal-regulated kinase (ERK), p38, and Jun N-terminal kinase (JNK) subfamilies. MAPK pathways act to regulate the expression of multiple genes that, together, regulate the inflammatory response [8]. Epigenetic processes interpret the genomic program in a cell-type and extracellular environment-dictated mode. Epigenetic information is encoded through covalent modifications of histones and DNA, nucleosome position, and substitution by histone variants. Recent technological advances have made it possible to assess different classes of epigenetic enzymes/factors, including components of the MAPK pathway [9], directly bound to genes that regulate transcription [2,3,4]. In parallel, an increasing number of small molecules are being developed that target epigenetic enzymes, mostly in oncology [10,11,12]. Many of these agents are in clinical trials [13]. 

Since the 1980s, several small molecules and biologics have been tested in clinical trials against sepsis. These treatments targeted established mediators and mechanisms driving septic physiology and included antibodies against lipopolysaccharide (LPS), corticosteroids, TLR4 and IL-1β antagonists, anti-TNF agents, and drugs targeting the coagulation cascade [14]. However, the intense research efforts and following clinical trials have not delivered specific treatments for this devastating syndrome. The reasons for failures stem from the incompletely understood pathophysiology of sepsis, the use of inadequate preclinical models, as well as the inability to tailor the treatment to selected patient groups to specific therapies [14]. An alternative approach to overcome the lack of progress is to repurpose existing oncology drugs for sepsis treatment. In this regard, the examples include topoisomerase I inhibitors [15], MEK1/2 [16], and ALK [17] kinases inhibitors, or immune checkpoint inhibitors [18,19,20]. 

Here, we performed the screening of several small molecules targeting chromatin-associated enzymes, including kinases, to assess their efficacy to inhibit TNFa production in vitro. Based on the in vitro results, we selected SCH772984, an inhibitor of ERK1/2, for in vivo studies. SCH772984 significantly improved survival in a sepsis mouse model induced by a lethal dose of LPS and a sepsis model induced by cecal ligation and puncture (CLP). RNA-seq transcriptomic signatures of SCH772984 treatment point to the attenuation of pathways associated with sepsis, providing, at least in part, molecular insights about this agent’s beneficial effects.

## 2. Results

### 2.1. ERK1/2 Inhibitor (SCH772984) Significantly Attenuates LPS-Induced TNFa Release and Inflammatory Genes Expression in Murine Macrophages Cell Line RAW 264.7

To determine which epigenetic drugs could have anti-sepsis properties, we undertook a supervised screening of several drugs targeting representatives of different classes of epigenetic regulators that have already been tested in at least early phases of clinical trials (Table 1). We first evaluated cell line models that would be suitable for the screening of these agents. To this end, we tested four noncancerous murine cell lines that were derived from different organs, as surrogates of organs that undergo dysfunction in severe sepsis, including lung fibroblasts (MLg), kidney inner medullary-collecting duct cells (mIMCD-3), primary hepatocytes (AML12), and transformed macrophages (RAW 264.7). Only RAW 264.7 cell lines responded robustly to the LPS challenge with the TNFa production (Appendix A); therefore, this cell line was chosen for the initial drug screening. The assay was optimized to achieve Z values of 0.5 and 0.77 for 4 h and 24 h time points of LPS challenge, respectively, a cutoff that is regarded as acceptable for high-throughput screening assays [21]. For in vitro testing, we used the following small molecules: inhibitors of MAPKs; MEK1/2 (AZD6244) and ERK1/2 (SCH772984), inhibitors of histone deacetylases: (HDAC) 1 and 3 (Entinostat, MS-275), and SIRT1 (Selisistat; EX-527); inhibiotors of histone methyltransferases: DOTL1 and EZH2 (Pinometostat, EPZ5676, and Tazemetostat, EPZ6438); inhibitors of histone demethylase: LSD1 (ORY-1001) and blockers of BRD proteins (RVX-208, OTX015) (Table 1). 

JQ1 inhibitor was included as a positive control as it has already been established as a potent inhibitor of macrophages’ inflammatory responses [31]. The efficacy of these agents to block TNFa production was assayed 4 h and 24 h post-LPS challenge by measuring half-maximal inhibitory concentration (IC_50_) values. The most potent inhibitors were OTX015, AZD6244, and SCH772984, with IC_50_ values of 0.31 µM, 0.33 µM, and 0.44 µM, respectively, at 24 h post-LPS challenge (Figure 1A). The inhibitors targeting other classes of epigenetic proteins did not block TNFa release, with the exception of MS-275 (IC_50_ of 1.54 µM). As the inhibitors of BET proteins [10] and MEK1/2 kinase [16] have already been shown to ameliorate sepsis in in vivo models, and no such studies were performed with ERK inhibitors, we focused on SCH772984 that inhibited TNFa production in vitro. The SCH772984 treatment of RAW 264.7 cells significantly blocked ERK1/2 phosphorylation at 1 h and 4 h LPS challenge (Figure 1B) and significantly reduced the mRNA expression of inflammatory genes, including Tnf, Ccl2, IL-6, and Lcn2, at least at a single time point from 1 h, 4 h, and 24 h, following LPS challenge (Figure 1C).

### 2.2. SCH772984 Changes RAW 264.7 Cells Transcriptomes by Altering Gene Expression Pathways Related to Immune System Pathways upon LPS Challenge

To elucidate the molecular mechanism underlying the in vitro SCH772984 blocking of TNFa production and ameliorating the immune response in macrophages, we conducted RNA-Seq of quiescent (0 h) and LPS-challenged RAW 264.7 cells (1 h, 4 h, and 24 h) treated with SCH772984. The pair-wise analyses revealed 35, 343, 1661, and 4123 differentially expressed genes (DEGs) (with adj. *p*-value < 0.05) with SCH772984 treatment at 0 h, 1 h, 4 h, and 24 h LPS time points, respectively (Appendix A). Of these, 35/0, 157/186, 937/724, and 1924/2199 DEGs were down-/up-regulated at 0 h, 1 h, 4 h, and 24 h time points, respectively, and 91/29 down-/up-regulated transcripts were shared between LPS time points (Figure 2A). To uncover functional changes caused by the SCH772984 at a given time point of LPS challenge, we performed pathway overrepresentation analysis of significantly up and downregulated DEGs (an adjusted *p*-value < 0.05 and fold change; FC ≥ 1.5) using the Reactome database. The functional analysis yielded 0/0, 2/0, 7/23, and 0/15 significantly (q-value < 0.05) enriched pathways for down-/up-regulated DEGs in 0 h, 1 h, 4 h, and 24 h, respectively (Appendix A and Figure 2B).

Significantly, Cytokine Signaling and Immune system Signaling by Interleukins were the only two Reactome pathways that included downregulated genes at 1 h and 4 h time points in SCH772984-treated cells. For the upregulated genes, there were seven Reactome pathways shared between 4 h and 24 h time points of SCH772984 treatment: GPCR and Interferon Signaling pathways. Overall, RNA-seq analysis revealed an increasing number of DEGs post-LPS/ SCH772984 challenge and uncovered both the negative and positive regulation of immune system pathways by SCH772984 in the LPS-treated RAW 264.7 cell line.

### 2.3. The SCH772984 Does Not Influence Chromatin Accessibility Genome-Wide in RAW 264.7 Cells Treated with LPS

To better understand the regulatory specificity of expression changes with SCH772984 treatment of RAW 264.7 cells, we profiled the chromatin accessibility dynamics using the ATAC-Seq method. To this end, we employed the same experimental setup as in the RNA-Seq survey with two biological replicates per time-point and treatment, as recommended by the ENCODE consortium [32]. On average, there were 113M reads mapped for the ATAC-Seq library. To statistically test for the differentially accessible regions (DARs), we compared normalized read counts within the consensus peaks called with MACS2. However, for the SCH772984 treatment upon the LPS challenge, there were no significant DARs at any LPS time point at the corrected *p*-value < 0.05. On the other hand, for the pair-wise comparison of quiescent vs. LPS-challenged cells, there were 268, 8066, and 596 DARs (corrected *p*-value < 0.05) at 1 h, 4 h, and 24 h time-points, respectively (Appendix A). The TNF gene was among the top ten genes at the 1 h time-point of the LPS challenge with three significant DARs at this locus. The inspection of the TNF locus at the genomic browser indicated a robust increase in the chromatin accessibility at 1 h and 4 h, followed by the chromatin closure at 24 h of LPS challenge (Appendix A). Overall, the ATAC-Seq survey did not show the influence of SCH772984 on chromatin accessibility that would explain its impact on transcriptome changes and the suppression of inflammatory response in RAW 264.7 cells. These results suggest that LPS alters chromatin accessibility independent of ERK1/2.

### 2.4. SCH772984 Significantly Improves Survival in Mouse Models of Sepsis and Reduces Plasma Levels of Ccl2/Mcp1 Chemokine

As SCH772984 significantly suppressed the expression of key inflammatory genes in vitro, this prompted us to test its potency in in vivo models of sepsis. The treatment of mice with 10 mg/kg of SCH772984 2 h following the injection of the LPS lethal dose (20 mg/kg) with repetitive SCH772984 dosing every 6 h significantly improved mice survival (Figure 3A). SCH772984 also significantly improved survival in a CLP-induced sepsis mouse model (Figure 3B). Given the protective effect of SCH772984 on mice survival, we next repeated the experiment using the CLP model with the 6 and 12 h endpoint and surveyed plasma levels of 26 cytokines/chemokines in CLP, SCH772984-treated CLP, and sham-operated animals, hereby referred to as the CLP, CLP+ SCH772984, and sham groups, respectively. Out of 26 proteins levels evaluated in plasma, the abundances of seven were significantly altered in the CLP+ SCH772984 group, when compared to the CLP, at least at the one time point. Specifically, the SCH772984 treatment significantly (*p*-value < 0.05) inhibited the Ccl2/MCP-1 plasma level at the 12 h time point. In contrast, the plasma levels of IL-2, IP-10 (CXCL10), MIP-1α, and IL-4, IL-27 were significantly elevated at 12 h and 6 h time points post-CLP, respectively, in the CLP+SCH772984 group. Finally, MIP-1β levels were significantly higher upon SCH772984 administration at the 6 h and 12 h time points (Figure 3C). The levels of other inflammatory mediators most relevant in sepsis, including TNFa, Il-6, and Il-1b, while increased in our model of CLP, remained unchanged upon SCH772984 treatment (Appendix A). Additionally, there were no differences in ERK1/2 phosphorylation in the liver and lung tissues assayed in three random animals per treatment following SCH772984 administration (Appendix A). Together, these results indicate that ERK1/2 blockage with a specific inhibitor improves the survival of animals in two models of sepsis and significantly dampens Ccl2 serum levels in a course of sepsis.

### 2.5. SCH772984 Suppresses Molecular Processes Associated with the Immune Response and Hemostasis in the Kidney and Liver and Activates Extracellular Matrix (ECM) Organization and Retinoic Acid (RA) Signaling Pathways in the Lungs and Liver in a CLP-Induced Sepsis Mouse Model

To characterize the molecular mechanism underlying the amelioration of CLP-induced sepsis by the SCH772984, we characterized the transcriptomes by RNA-Seq in the lung, kidney, liver, and heart tissues collected from four animals in each CLP, CLP+ SCH772984, and sham group at 6 h and 12 h time points post-CLP. In summary, 80 organs were subjected to the transcriptome survey, and sets of organs were from the same animals in a given group. First, we wished to determine the extent of transcriptome changes and the enrichment of cellular pathways in our CLP model across four organs. The pair-wise comparison between the CLP and sham-operated mice groups identified 631, 2917, 4775, and 3867 DEGs (an adjusted *p*-value < 0.05) in the heart, kidney, lung, and livers samples at 6 h, and 1503, 6998, 6419, and 6310 DEGs at 12 h, respectively (Figure 4A; Appendix A). There were 58 (at 6 h) and 226 (at 12 h) DEGs with the same direction of change common to all organs (Figure 4A). The comparison of CLP and CLP+ SCH772984 mice groups generated 104, 62, 29, and 129 DEGs in the heart, kidney, lung, and livers, respectively, at 6 h, and 62, 3225, 878, and 1673 DEGs, in the heart, kidney, lung, and livers, respectively, at 12 h (Figure 4B; Appendix A). There were only six DEGs downregulated shared among the organs at the 12 h time point.

To translate transcriptional information to the changes in molecular pathways, we performed functional analyses of significantly up- and downregulated DEGs for the sham vs. CLP and the CLP and CLP+ SCH772984 comparisons across the tissues (an adjusted *p*-value < 0.05 and fold change; FC ≥ 1.5) using the Reactome database. We focused on the pathways that were recurrently altered in more than two types of samples, with a differential gene ratio higher than 10%, which yielded 38 pathways that fall into the Immune system, Hemostasis, ECM organization, Metabolism, Signal Transduction, and Developmental Biology Reactome pathways parental categories (Figure 4C). 

The signature of inflammatory processes under the Immune system category was present in all organs, except for the heart, at least at one time point in CLP-operated animals. The SCH772984 treatment downregulated transcripts that were overrepresented in the several pathways under the Immune system category at the 12 h time point in the kidney and liver. Furthermore, the SCH772984 suppression of DEGs related to the Hemostasis-associated pathways was observed at the same time point in the kidney, lung, and liver tissues. The ECM organization master term encompassed downregulated DEGs in the lungs and heart at 6 h and 12 h time points and upregulated DEGs in the liver at 12 h in the CLP group. On the other hand, the SCH772984 treatment induced transcripts encompassing several ECM pathways in the lung tissue at 12 h post-CLP procedure. The signal transduction was another master term that contained significant pathways with the opposite direction of activity upon SCH772984 treatment. Specifically, the activation of RA signaling was observed in the liver at 12 h post-CLP upon SCH772984 treatment. Taken together, the transcriptome profiling of multiple organs following SCH772984 treatment in a CLP-induced mouse model of sepsis points to significant suppression of molecular processes associated with the immune response and hemostasis in the kidney and liver, as well as the activation of the ECM and RA signaling pathways in the lungs and liver, respectively. These SCH772984-induced gene expression effects could account, at least in part, for this agent’s improved survival properties in our model of experimental sepsis.

## 3. Discussion

Sepsis is defined as a life-threatening organ dysfunction caused by dysregulated host responses to infection [4]. While the treatment of sepsis relies on the treatment of the underlying infection and critical care, nearly a fifth of the 49 million cases of sepsis in 2017 were fatal [2]. Moreover, according to a World Health Organization epidemiology report, the true burden of sepsis is likely underestimated given that only 15% to 50% of patients with sepsis are correctly coded using the ICD system [25]. Unfortunately, despite multiple clinical trials [14,33], specific pharmacotherapies (other than antibiotics) for sepsis have not yet been implemented. 

Herein, using an in vitro screening of several compounds targeting chromatin-associated enzymes, along with survival data from in vivo murine sepsis models, we provide evidence that the SCH772984 agent is a promising small-molecule candidate for further clinical evaluation in the treatment of sepsis. We and others have shown before that components of the MAPK pathway [34], ERKs [35], and JNKs [36] can be found in the nucleus tethered to the chromatin at transcribed genes [34]. Therefore, we reasoned to include MAPK inhibitors, namely MEK1/2 (AZD6244) and ERK1/2 (SCH772984), in our initial in vitro screening of drugs comprising compounds targeting different classes of epigenetic enzymes. We included compounds tested at least in phase 1 of clinical trials (mainly oncology), and as such, indicated that these had already undergone preclinical and toxicology testing in animals. 

We chose the heart, kidney, lung, and liver tissues for the RNA-Seq survey upon SCH772984 treatment as those organs are frequently affected in the course of sepsis [37]. Our recent RNA-seq transcriptomic survey of the same tissues 6 h, 12 h, and 24 h post-CLP procedure in mice indicated that the alteration of the immune system pathways is common across multiple organs, and the highest number of significant DEGs in those organs, when compared to sham-operated mice, is observed at 12 h [38]. Based on this observation and the fact that CLP mice start manifesting clinical signs of sepsis at around 12 h post-CLP [39], we selected the 6 h and 12 h time points to capture early molecular events of SCH772984 action. Previous studies have shown that for some therapeutic interventions, protective effects in mid-grade sepsis did not work for a more severe form of sepsis [40]. Thus, we used a high severity grade of CLP sepsis to ascertain if SCH772984 intervention could be an effective strategy to ameliorate sepsis-associated death. 

With CLP, we identified the following SCH772984 gene expression effects. (i) Alteration of the immune system pathways in the kidney and liver. (ii) Overrepresentation of downregulated transcripts involved in hemostasis and platelet activation, signaling, and aggregation pathways. In this regard, coagulopathy is a hallmark of sepsis, manifesting liver dysfunction [41], and platelets dysfuncion known in septic organ dysfunction [42]. (iii) Upregulation of the ECM pathways only in the lungs and upregulation of the RA signaling only in the liver. It is known that ECM plays an important role in bacterial infection and undergoes a vast remodeling both as a result of its degradation by pathogens releasing proteases and the local immune cells response to the infection [43], while RA is a potent inducer of liver regeneration [44] and a modulator of the immune responses [45].

SCH772984 is an ATP-competitive inhibitor that binds to the unphosphorylated, inactive form of ERK1/2, and potently and selectively inhibits ERK1 and ERK2 activity, with IC_50_ values of 4 and 1 nM, respectively [23]. Several studies have shown that SCH772984 inhibited BRAF/RAS-mutant cancer cells proliferation and increased apoptosis in vitro and in vivo [23,46,47,48]. Furthermore, a combination of SCH772984 with other anticancer drugs, including Cucurbitacin B [49] and dabrafenib [50], had a synergistic effect on tumor growth inhibition in pancreatic and thyroid cancer in vivo models, respectively. Apart from oncology, Wong and colleagues have used a combination of siRNA knockdown and SCH772984 to define the role of LPS-induced ERK activation in primary human endothelial cells. Contrary to the ERK1/2 activation promoting IL-6 production in monocytes, blunting the ERK1/2 activity increased IL-6 production in these endothelial cells [51]. 

Repurposing the large arsenal of existing cancer drugs is an attractive proposition to expand the clinical pipelines for sepsis treatment. A study by Rialdi et al. showed that the topoisomerase I inhibitors, approved for ovarian, lung, and colon cancer treatment, can significantly prolong survival in an animal model of sepsis by attenuating an acute inflammatory reaction in a mechanism that blocks the RNA polymerase II transcriptional activity at a set of genes that require chromatin remodeling and are activated immediately by immune cells during pathological inflammation [15]. Another study by Smith et al. tested Trametinib, a MEK1/2 inhibitor, approved for unresectable or metastatic melanoma with BRAF^V600^ mutation, in a CLP mouse model of sepsis [16]. Although the study did not present survival data, Trametinib treatment reduced hypothermia, serum proinflammatory cytokines, and improved levels of liver and renal tubular injury markers. On a molecular level, these observations were accompanied by the reduction in ERK kinase activation and decrease in mRNA expression of inflammatory mediators, including TNFα, IL-1β, and IL-6 in the renal cortex of trametinib-treated CLP mice [16]. More recently, the ALK kinase inhibitor LDK378, the FDA-approved oral anticancer drug, was shown to have anti-inflammatory activity in animal models of lethal sepsis [17]. Zeng et al. reported that the pharmacologic or genetic suppression of ALK inhibited the activator of interferon genes (STING) pathway activation in monocytes and macrophages. Furthermore, the ALK KO mice were more resistant to polymicrobial sepsis. This improvement was reflected in the biochemical measurement of creatine kinase, alanine aminotransferase, blood urea nitrogen, and amylase in the heart, liver, kidney, and pancreas tissue, respectively. Overall, these findings support therapeutic role for ALK in lethal infection and open a venue for further clinical testing. Blocking immune checkpoints with monoclonal antibodies is regarded as a breakthrough in cancer therapy. Among the immune checkpoint inhibitors, PD-1/PD-L1 and CTLA-4 inhibitors significantly improved survival and clinical manifestation in preclinical models of sepsis [18,19,20], which lead to their testing in a human clinical trial of severe sepsis [20]. Recently, we have shown that MNK1/2 inhibitors, initially developed for cancer treatment, could be useful as a novel immunomodulatory therapy of inflammatory diseases, including severe sepsis [52]. In summary, these examples underscore that clinically approved drugs, or those under development for oncology, could be repurposed for the treatment of severe sepsis. 

## 4. Materials and Methods

### 4.1. Cell Line Models for In Vitro Studies

mIMCD-3 (CRL-2123), AML12, (CRL-2254), MLg (CCL-206), and RAW 264.7 (TIB-71) mouse cell lines were purchased from the ATCC (Manassas, VA, USA) and were maintained under the recommended conditions. 

### 4.2. Screening of Compounds for Inhibition of TNFa Production

For AlphaLISA, 10000 cells/well were seeded on a 384-well plate (#6005350, Perkin Elmer, Waltham, MA, USA) using the MultiFlo FX Microplate Dispenser. The next day, cells were treated with LPS (L9143-02, Sigma-Aldrich, Saint Louis, MO, USA) at a final concentration of 10 µg/mL, the plates were then incubated at 37 °C for 4 and 24 h, and the TNFa concentration was measured using AlphaLISA (Perkin Elmer, Waltham, MA, USA) on the Spark reader (Tecan, Männedorf, Switzerland) and the AlphaScreen module as per the manufacturer’s protocol. The CellTiter-Glo Luminescent Cell Viability Assay (Promega, Madison, WI, USA) microplate assay was used to measure cell viability. Before the LPS challenge, cells were pre-treated for 1 h with the following drugs purchased from Selleckchem: AZD6244, SCH772984, MS-275, EX527, EPZ-5676, EPZ-6438, ORY-1001, RVX-208, OTX015, and (+)-JQ1 targeting the chromatin-associated proteins MEK 1/2, ERK 1/2, HDAC 1/2/3, SIRT1, DOT1L, EZH2, LSD1, BRD 2/3/4, BRD2/3, and BRD4 (Selleckchem, Houston, TX, USA), respectively, at the concentrations of 5, 1, 0.2, 0.04, 0.008, 0.0018, and 0.00032 µM. Drugs were dispensed using the Microlab STAR unit (Hamilton, Reno, NV, USA), while the transfer and media dilutions for the AlphaScreen readout were performed with the CyBio SELMA liquid handler (Analytic Jena, Jena, Germany). Measurements were performed in 12-plicate.

### 4.3. Assessing the Compound SCH772984 against LPS Challenge in RAW 264.7 Cell Line

Here, 5 × 10^5^ RAW 264.7 cells were seeded in the presence of 10% Fetal Bovine Serum (FBS) on a 6-well plate a day before the treatment. The next day, cells were pretreated with 500 nM of SCH772984 for 1 h and then challenged with 10 µg/mL of LPS for 1, 4, and 24 h in the presence of 500 nM of SCH772984. Cells were collected at indicated time points and pellets were frozen at −80 °C for later RNA isolation or immediately used for the ATAC-Seq protocol.

### 4.4. RNA Isolation

Total RNA was isolated from RAW 264.7 cells and tissues using TRIzol reagents (T9424, Sigma-Aldrich, Saint Louis, MO, USA). Then, 3 mm^3^ tissue fragments were cut off in a dry ice-chilled workplace. Fragments were grinded with a disposable pestle (749520-0590, DWK, Mainz, Germany) in 200 µL of TRI Reagent and left to dissolve further for 2 min in RT. To remove debris, samples were centrifuged at 14,000× *g* for 30 s. The supernatant containing RNA was further processed using the Direct-zol^TM^ RNA MiniPrep Plus kit (R2072, Zymo Research, Irvine, CA, USA) following manufacturer instructions. RNA was stored at −80 °C.

### 4.5. Reverse Transcription Quantitative (RT-q)PCR

The RT reaction was performed using the High-Capacity cDNA Reverse Transcription Kit (4368814, Thermo Fisher Scientific, Waltham, MA, USA) with 500 ng of total RNA according to the manufacturer’s protocol. qPCRs of cDNA were carried out using TaqMan^®^ Gene Expression Assays and the SensiFAST™ Probe Hi-ROX Kit (BIO-82005, Meridian, Memphis, TN, USA) on the Applied Biosystems 7900HT Fast Real-Time PCR System, as described in [53]. The following TaqMan Gene Expression Assays (Thermo Fisher Scientific, Waltham, MA, USA) were used: Tnf (Mm00443258_m1), Ccl2 (Mm004412142_m1), Il6 (Mm00446190_m1), Lcn2 (Mm01324470_m1), Hprt1 (Mm00446968_m1), and Gapdh (Mm99999915_g1). RT-qPCR data were normalized to the geometric mean expression of Hprt1 and Gapdh mRNAs using the ∆∆Ct method. 

### 4.6. RNA-Seq Data Analyses

RNAseq cDNA libraries were prepared with the TruSeq Stranded Total RNA Library Prep Kit with the Ribo-Zero Human/Mouse/Rat Sample Prep Kit (#RS-122-2203, Illumina, San Diego, CA, USA) at the CeGaT GmbH (Tübingen, Germany) and sequenced there with the Illumina NovaSeq 6000 Platform using 100 bps paired-end reads. Raw sequences were trimmed according to quality using Trimmomatic (version 0.39) [54] using default parameters, except MINLEN, which was set to 36. Trimmed sequences were mapped to Mus musculus, version mm10, a reference genome provided by Ensembl, using Hisat2 [55] with default parameters. Optical duplicates were removed using the MarkDuplicates tool from the GATK [56] package (version 4.1.2.0) with default parameters, except the Optical Duplicate Pixel Distance parameter set to 12000. Mapped reads were associated with transcripts defined in the Ensembl database [57] (GRCm38, version 95) using HTSeq-count (version 0.9.1) [58] with default parameters, except stranded set to “reverse.” Differentially expressed genes were selected using the DESeq2 package [59] (version 1.16.1). 

#### Functional Analyses of the Transcriptomic Data

The overrepresentation of Reactome pathways in genes that were differentially expressed was performed with R package ClusterProfiler version 3.6, separately for up- and downregulated genes [60]. A heatmap visualization was conducted with ComplexHeatmap R package version 2.3.1, using only pathways present in at least two data points (for LPS treatment) and with a ratio of at least 0.1 of differential vs. nondifferential genes present in a pathway. For clarity, only the top 3 pathways from each Reactome category for each data point were taken into account. 

### 4.7. Assay for Transposase-Accessible Chromatin Using Sequencing (ATAC-Seq)

The ATAC-Seq was performed according to the Buenrostro et al. protocol [61] with minor modifications. In addition, 5 × 10^5^ RAW 264.7 cells were washed using 50 μL of ice-cold 1x PBS and centrifuged at 500× *g* for 7 min at 4 °C. To obtain nuclear pellets, cells were lysed using 50 μL of cold lysis buffer (10 mM of Tris-HCl, pH 7.4; 10 mM of NaCl; 3 mM of MgCl_2_; 0.1% IGEPAL CA-630) and immediately spun at 500× *g* for 10 min at 4 °C. To avoid losing cells/nuclei, we carefully pipetted away the supernatant from the pellet. The nuclei pellets were resuspended in 50 μL of ice-cold transposition reaction mix: 25 μL of 2x TD buffer (20034211, Illumina, San Diego, CA, USA); 2.5 μL of TDE1 (Nextera Tn5 Transposase, 20034211, Illumina, San Diego, CA, USA); and 22.5 μL of nuclease-free water. The transposition reaction was carried out for 30 min at 37 °C with 300 rpm mixing. Directly following transposition, the samples were purified using the DNA Clean & Concentrator kit (D4013, Zymo Research, Irvine, CA, USA) according to the manufacturer’s protocol. Libraries were constructed using: 10 μL of Transposed DNA; 10 μL of nuclease-free water; 25 μL of NEBNext^®^ Ultra™ II Q5^®^ Master Mix (M0544, NEB, Ipswich, England), and 2.5 μL of 25 μM of custom Nextera PCR primer 1 (i5) and primer 2 (i7) (Invitrogen, Waltham, MA, USA), using the following PCR conditions: 72 °C for 5 min, 98 °C for 30 s—1 cycle, followed by thermocycling at 98 °C for 10 s, 63 °C for 30 s, and 72 °C for 1 min—5 cycles. To reduce the GC and size bias, we monitored the PCR reaction using qPCR to stop amplification before saturation. To this end, after the 5 cycles, a volume of 5 μL of the PCR reaction was mixed with: 4.1 μL of nuclease-free water; 0.25 μL of 25 μM of custom Nextera PCR primer 1 (i5) and primer 2 (i7); 0.1 μL of 100x SYBR Green I; and 5 μL of NEBNext^®^ Ultra™ II Q5^®^ Master Mix. The qPCR reaction was run under the following conditions: 98 °C for 30 s—1 cycle, followed by thermocycling at 98 °C for 10 s, 63 °C for 30 s, and 72 °C for 1 min—20 cycles. To calculate the additional number of cycles needed for the remaining 45 μL reaction, we analyzed the plotted linear Rn versus cycle and determined the cycle number that corresponds to 1/3 of the maximum fluorescent intensity. Libraries were amplified for a total of 16 cycles and purified using the DNA Clean & Concentrator kit (D4013, Zymo Research, Irvine, CA, USA) according to the manufacturer’s protocol. Library quality and concentration were determined by the Agilent High Sensitivity DNA Kit on a Bioanalyzer 2100 (Agilent; 5067-4626, Agilent, Santa Clara, CA, USA). ATAC-Seq libraries were sequenced with the Illumina NovaSeq 6000 Platform with 75 bps paired-end reads at CeGaT GmbH (Tübingen, Germany). Two biological replicates were analyzed per treatment. ATAC-Seq reads were processed using Trimmomatic (version 0.39) using default parameters and then aligned to the mouse reference genome mm10 using bowtie2 [62] (version 2.2.6). Duplicate reads were marked using Picard (version 2.24), and reads with MAPQ < 30 and in the Encode blacklist regions were excluded with SAMtools [63] (version 1.7). Peaks were called with MACS2 (version 2.1.1). Differentially Accessible Regions (DARs) were determined by DiffBind (version 3.0.12).

### 4.8. Western Blot

Cells and tissues were lysed in RIPA buffer (1% Triton X-100, 0.5% sodium deoxycholate, 0.1% SDS, 50 mM of Tris pH 7.4, 150 mM of NaCl, and 0.5 mM of EDTA) supplemented with a cocktail of protease and phosphatase inhibitors (78441, Thermo Fisher Scientific, Waltham, MA, USA). Tissues were sonicated in a Bioruptor Plus (Diagenode, Philadelphia, PA, USA) using a 30 s on–off cycles protocol for 10 min at high intensity followed by centrifugation at 12,000× *g*, 4 °C, and 10 min. Protein concentration was measured with a BCA Protein Assay Kit (23225, Thermo Fisher Scientific, Waltham, MA, USA). Then, 20 µg of total protein per sample was resolved on 10% SDS-PAGE and transferred to the PVDF membrane (IPVH00010, Merck Millipore, Burlington, MA, USA). Membranes were blocked in 5% Bovine Serum Albumin (BSA) in Tris-Buffered Saline, with Tween 20 (TBST) followed by incubation with specific primary and secondary antibodies. For signal detection, the Clarity Western ECL Substrate (170-5061, Bio-Rad, Hercules, CA, USA) and ChemiDoc imaging system (Bio-Rad, Hercules, CA, USA) were used. Antibodies used: pErk1/2 (Thr202/Tyr204) (4370, Cell Signaling Technology, Danvers, MA, USA) and Actb (ab8226, Abcam, Cambridge, UK). The level of ERK1/2 phosphorylation was quantified by densitometric analysis using ImageJ software. For each blot, the background-subtracted density of the protein band was divided by the density of the loading control. The mean and standard deviations for the biological replicates were determined.

### 4.9. Assessing the Effect of Compound SCH772984 in Mouse Models

#### 4.9.1. Mice

C57BL/6W (own breeding) mice males, 16–20 weeks of age, were maintained under the specific pathogen-free animal facility at the Maria Skłodowska-Curie National Research Institute of Oncology according to the institutional guidelines.

#### 4.9.2. LPS-Induced Septic Shock in Mice

Weight-matched 16–20 weeks-old male mice were injected with 20 mg/kg of LPS into the contralateral side of the abdomen.

#### 4.9.3. Cecal Ligation and Puncture (CLP)-Induced Septic Shock in Mice

The CLP procedure to induce high-grade sepsis was performed as previously described [39]. Briefly, under isoflurane (Forane) anesthesia, mice were laid supine and a midline incision was made in the lower abdomen after shaving and sterilizing the abdominal skin. The cecum was identified and extracted. Stool in the cecum was squeezed distally and a 3–0 suture was tied proximal to the base of the cecum. A 21 g needle was used to make a through-and-through cecal puncture, and a small amount of stool was extracted. The cecum was placed back in the abdominal cavity and the abdominal wall was closed with 6–0 sutures. Immediately post-op, mice were given 20 mL/kg of normal saline solution and buprenorphine, 0.05 mg/kg. Every 6 h, mice were dosed with normal saline solution, 10 mL/kg, and buprenorphine, 0.05 mg/kg.

#### 4.9.4. Mice Monitoring Following Inducing Septic Shock

Mice were maintained at room temperature in cages, 4 individuals each, with access to fresh water and food throughout the experiment. Mice were monitored by two investigators every 2 h after the induction of sepsis. The severity of sepsis was assessed by five variables and the 0–4 scoring system, including physical appearance, body weight, measurable clinical symptoms, unprovoked behavior, and response to external stimuli (Appendix A). Mice were euthanized whenever two of the five observed parameters occurred on a scale of 3 or if the total score of the observed parameters was 8. 

#### 4.9.5. SCH772984 Administration to Mice

The SCH772984 at 10 mg/kg or formulation (3% DMSO + 10% polyethylene glycol) was injected *i.p.* 2 h following LPS (20 mg/kg) dosing in the survival experiment, and then every 6 h later, the SCH772984 was administrated. For the 6 h and 12 h time points and survival, CLP experiments’ mice were injected *i.p.* with 10 mg/kg of SCH772984 or formulation 2 h following the CLP procedure and every 6 h with another dose of SCH772984 or formulation. In summary, mice at 6 h and 12 h time points received a single and double dose of SCH772984, respectively. At 6 h and 12 h post-CLP, anesthesia was induced with isoflurane and blood was collected via cardiac puncture. The chest was opened and the heart and lungs collected rapidly into cryovials, which were immediately snap-frozen in liquid nitrogen. The kidney and liver were collected similarly immediately thereafter. 

### 4.10. Cytokine Screening and Analysis with 26-Plex Mouse ProcartaPlex

Cytokine & Chemokine 26-Plex Mouse ProcartaPlex™ (EPXR26026088901, Thermo Fisher Scientific, Waltham, MA, USA) (GM-CSF, IFN-γ, IL-1β, IL-2, IL-4, IL-5, IL-6, IL-12p70, IL-13, IL-18, TNF-α, IL-9, IL-10, IL-17A [CTLA-8], IL-22, IL-23, IL-27, Eotaxin [CCL11], GRO alpha [CXCL1], IP-10 [CXCL10], MCP-1 [CCL2], MCP-3 [CCL7], MIP-1 alpha [CCL3], MIP-1 beta [CCL4], and MIP-2 RANTES [CCL5]) was used to screen cytokines in 50 μL of plasma according to the manufacturer’s instructions.

## 5. Conclusions

Our study identified the ERK1/2 inhibitor, the SCH772984 compound, as a promising drug improving survival in a preclinical model of sepsis. Transcriptomic signatures of SCH772984 compound action in vivo highlighted its influence on immune response, the platelet-related signaling, ECM, and RA signaling pathways. Recently, to improve the pharmacokinetics of SCH772984, the medical chemistry optimization yielded the MK-8353 compound [64] that was tested in phase 1 clinical trials in cancer patients [65]. Given that this improved version of SCH772984 is currently tested in oncology clinical trials, our study indicated that this SCH772984 successor could be considered for the treatment of severe sepsis, as well. In summary, this and other studies underscore that not only can cancer drugs be used to ameliorate sepsis, but as in oncology, a combination of such agents could further improve clinical outcomes in sepsis. 

## Figures and Tables

**Figure 1 ijms-22-10204-f001:**
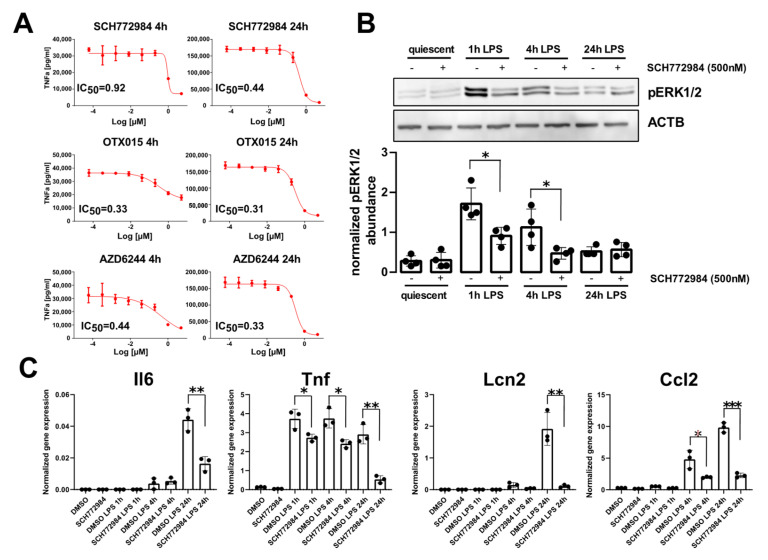
SCH772984 significantly attenuates LPS-induced TNFa production and inflammatory genes expression in the RAW 264.7 cell line. (**A**) IC_50_ values of TNFa production inhibition for the most effective drugs at 4 h and 24 h post-LPS challenge. (**B**) Representative Western blot (upper panel) showing the levels of ERK1/2 phosphorylation following SCH772984 treatment upon LPS time course challenge. The graph (lower panel) shows the averaged pERK1/2 abundance normalized to Actb by densitometry analysis of Western blotting bands. Values derived from four independent experiments and their means (+/− SD) are presented. * *p* ≤ 0.05, as determined by unpaired *t*-test. (**C**) qPCR results showing the mRNA expression of inflammatory genes following SCH772984 treatment upon LPS time course. Data from three independent experiments and their means and +/− SD are presented. * *p* ≤ 0.05, ** *p* < 0.01, *** *p* < 0.001 as determined by unpaired *t*-test.

**Figure 2 ijms-22-10204-f002:**
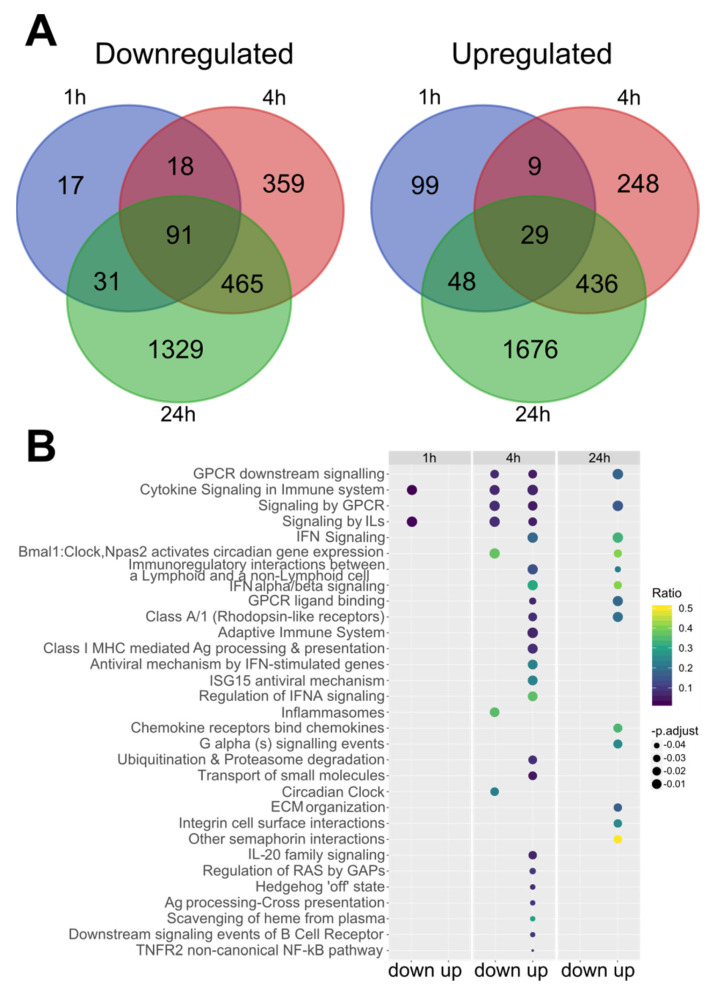
SCH772984 changes RAW 264.7 cells transcriptomes by altering gene expression pathways related to immune system pathways upon LPS challenge. (**A**) Venn diagram illustrating the number of up and downregulated DEGs (adj. *p*. value < 0.05) and Reactome pathways (**B**) altered upon SCH772984 treatment during the LPS challenge time points. IL—interleukin; IFN—interferon; Ag—antigen; ECM—extracellular matrix.

**Figure 3 ijms-22-10204-f003:**
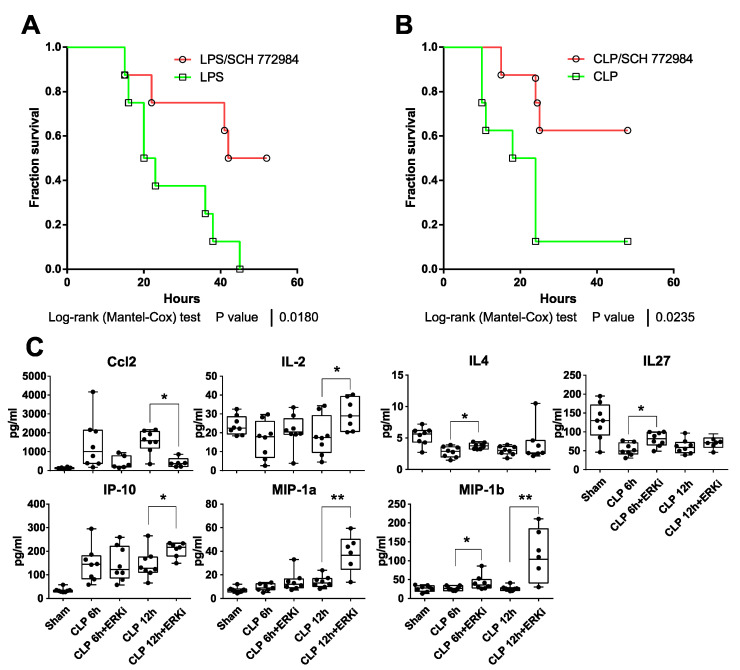
SCH772984 improves survival in mouse models of sepsis and reduces plasma levels of Ccl2/Mcp1 chemokine. Kaplan–Meier survival curves of (**A**) LPS-treated C57BL/6W mice (20 mg/kg, i.p., *n* = 8 per group) or mice subjected to cecal ligation puncture (CLP) procedure (**B**) that were both injected i.p. with solvent control or 10 mg/kg of SCH772984 2 h after LPS administration/CLP procedure. (**C**) Plasma levels of indicated cytokines were measured with the 26-Plex Mouse ProcartaPlex immunoassay in CLP, SCH772984 (ERKi)-treated CLP, and sham-operated animals (*n* ≥ 6) at 6 and 12 h endpoint. ** *p* < 0.01, * *p* < 0.05 as determined by unpaired *t*-test. Only seven cytokines with significant alternations in plasma levels following SCH772984 are shown.

**Figure 4 ijms-22-10204-f004:**
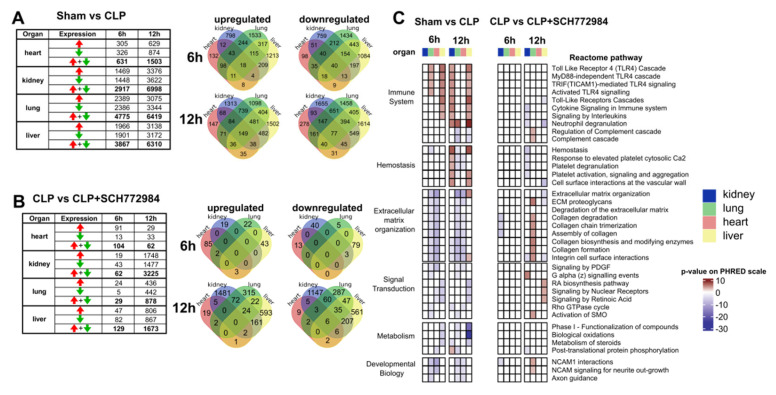
SCH772984 downregulates transcripts associated with the immune response and hemostasis in kidney and liver tissues, and positively regulates ECM and retinoic acid pathways in the lungs and liver, respectively, in a CLP-induced mouse model of sepsis. The numbers of differentially expressed genes (adj. *p*-value < 0.05) across the organs and two time points in the sham vs. CLP groups (**A**) and upon SCH772984 (10 mg/kg) treatment (**B**). Red and green arrows indicate up- and downregulated genes, respectively. Venn diagrams show the overlapping numbers of DEGs for organs at a given time point post-CLP for up-and downregulated genes. (**C**) Significantly altered (adj. *p*-value < 0.05) recurrent Reactome terms across the organs and time points in the sham vs. CLP and the CLP vs. CLP+ SCH772984 groups. The Reactome pathways with a differential gene ratio higher than 10% and recurrently altered in more than two types of samples are shown. The *p*-value is shown on the PHRED scale and is reversed for downregulated pathways.

**Table 1 ijms-22-10204-t001:** List of small molecules together with representative IDs of clinical trials and used in in vitro screening.

Host Protein	Compound Targeting the Host Protein	Mechanism of Action of the Compound with Relevant Reference	Examples of Clinical Trials	TNFa Production Inhibition 4/24 h Post-LPS Treatment in RAW 264.7 Cell Line (IC_50_ Values µM)
MEK1/2	AZD6244	non-ATP-competitive MEK1/2 kinase inhibitor [22]	NCT01635023	0.44/0.33
ERK1/2	SCH772984	selective and ATP competitive inhibitor of ERK1/2 [23]	NCT01358331	0.92/0.44
HDAC1/3	MS-275	Inhibitor of histone deacetylases (HDACs); induces autophagy and apoptosis in cancer cell lines [24]	NCT01594398	NA/1.54
SIRT1	EX-527	Potently inhibitory effect against SIRT1 HDAC activity [25]	NCT04184323	NA/NA
DOTL1	EPZ5676	an S-adenosyl methionine (SAM) competitive inhibitor of protein methyltransferase DOT1L [26]	NCT02141828	NA/NA
EZH2	EPZ6438	a potent, and selective of protein methyltransferase EZH2 inhibitor [27]	NCT02860286	NA/NA
KDM1A	ORY-1001	an orally active and selective lysine-specific demethylase KDM1A inhibitor [28]	2018-000482-36	NA/NA
BRD2	RVX-208	a potent BET bromodomain inhibitor for BD2, with 170-fold selectivity over BD1 [29]	NCT01423188	NA/1.56
BRD2/3/4	OTX015	a potent BET bromodomain inhibitor targeting BRD2, BRD3, and BRD4 proteins. Inhibits the expression of nuclear receptor binding SET domain protein 3 (NSD3) target genes [30]	NCT01713582	0.33/0.31
BRD4	(+)-JQ1	a BET bromodomain inhibitor, targeting BRD4(1/2), binding to all bromodomains of the BET family, but not to bromodomains outside the BET family [31]	Not in trials; stability issues in vivo	0.07/0.08

NA—not available.

## Data Availability

The RNA-Seq and ATAC-Seq data presented in this study are openly available in the Gene Expression Omnibus at identification numbers GSE179554 and GSE180085, respectively.

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
