# Peer review of "Selective Extracellular Signal-Regulated Kinase 1/2 (ERK1/2) Inhibition by the SCH772984 Compound Attenuates In Vitro and In Vivo Inflammatory Responses and Prolongs Survival in Murine Sepsis Models"

_ijms, 2021, doi:10.3390/ijms221910204_

Round 1

Reviewer 1 Report

In the paper “Selective Extracellular Signal-Regulated Kinase 1 /2 (ERK1/2) Inhibition by the SCH772984 Compound Attenuates in vitro and in vivo Inflammatory Responses and Prolongs Survival in Murine Sepsis Models” the authors have identified the SCH772984 compound, an extracellular signal-regulated kinase (ERK) 1/2 inhibitor, as effectively blocking TNF production in the lipopolysaccharide (LPS) stimulated mouse RAW264.7 macrophages. SCH772984 treatment improved survival in the LPS-induced lethal endotoxemia and in cecal ligation and puncture (CLP) mouse models of sepsis, and reduced plasma levels of Ccl2/Mcp1. Functional analyses of RNA-Seq data sets for kidney, lung, liver, and heart tissues from SCH772984 treated animals collected at 6h, and 12h post-CLP procedure revealed a significant overrepresentation of downregulated differentially expressed genes (DEGs) that are directly related to the immune response and platelets activation.

It is interesting preliminary study of early points in sepsis development which findings need further confirmation in the future studies.

Comments

  1. RAW264.7 is cancer cell line and data obtained from these cells serves well for author’s suggestion that “repurposing the large arsenal of existing cancer drugs is an attractive proposition to expand the clinical pipelines for sepsis treatment.” However, it is necessary to show similar response in non-cancer cells, as kidney cell line (mIMCD3), Mouse lung fibroblast cell line (MLg) and most importantly mouse liver cell line (AML12 cell line established from hepatocytes from a CD1 mouse strain transgenic for human TGF alpha) have not shown any TNF response upon LPS treatment. Any explanation? Also, TNF – which one or the assay is not specific for particular member of TNF family?
  2. Therefore, the authors have used cancer cell line to investigate SCH772984 treatment (with or without LPS) and for further screening and analysis. Data from the airway epithelial cells which represents a primary site for the entry of pathogenic bacteria into the lungs or kidney epithelial cells would be highly informative.
  3. Why is ERK 1/2 phosphorylation decreased in quiescent RAW264.7 upon the treatment with SCH772984 (Fig 1B)? How many samples/per group do you have for Western blot analysis? Could you show densitometry graph?
  4. It is very hard to read text (gene clusters) in the Figure 2B.
  5. What’s happening with TNF in mouse plasma LPS or CLP model after treatment with SCH772984?
  6. Western blot of pERK1/2 in mouse tissue (at least lung and liver tissue) should be shown.
  7. What is happening with the levels of inflammatory mediators most relevant in sepsis as TNFα, IL-1β, and IL-6 in LPS and CLP mouse model with and without treatment with SCH772984?
  8. Minor points
  9. Line 41-42, it’s stated: “outcome and determines transcriptome signatures of SCH772984 action in vitro and a preclinical model of sepsis.” I think it will be better to change in: … outcome and determines transcriptome signatures of SCH772984 action in vitro and in vivo, in a preclinical model of sepsis.
  10. Methodology should start with small molecule screening following by in vitro cell work and then in vivo mouse models.

Author Response

Rewiever#1

In the paper “Selective Extracellular Signal-Regulated Kinase 1 /2 (ERK1/2) Inhibition by the SCH772984 Compound Attenuates in vitro and in vivo Inflammatory Responses and Prolongs Survival in Murine Sepsis Models” the authors have identified the SCH772984 compound, an extracellular signal-regulated kinase (ERK) 1/2 inhibitor, as effectively blocking TNF production in the lipopolysaccharide (LPS) stimulated mouse RAW264.7 macrophages. SCH772984 treatment improved survival in the LPS-induced lethal endotoxemia and in cecal ligation and puncture (CLP) mouse models of sepsis, and reduced plasma levels of Ccl2/Mcp1. Functional analyses of RNA-Seq data sets for kidney, lung, liver, and heart tissues from SCH772984 treated animals collected at 6h, and 12h post-CLP procedure revealed a significant overrepresentation of downregulated differentially expressed genes (DEGs) that are directly related to the immune response and platelets activation.

It is interesting preliminary study of early points in sepsis development which findings need further confirmation in the future studies.

Comments:

  1. RAW264.7 is cancer cell line and data obtained from these cells serves well for author’s suggestion that “repurposing the large arsenal of existing cancer drugs is an attractive proposition to expand the clinical pipelines for sepsis treatment.” However, it is necessary to show similar response in non-cancer cells, as kidney cell line (mIMCD3), Mouse lung fibroblast cell line (MLg) and most importantly mouse liver cell line (AML12 cell line established from hepatocytes from a CD1 mouse strain transgenic for human TGF alpha) have not shown any TNF response upon LPS treatment. Any explanation?

Response: RAW264.7 is indeed a cancerous cell line as it was derived from Abelson leukemia virus-transformed macrophages from BALB/c mice. This cell line is however commonly used as a model of mouse macrophages for the study of cellular responses to microbes and their products (https://doi.org/10.1371/journal.pone.0198943). In our hands it was the only the one, from four tested, that effectively produced TNFa to the culture medium therefore was used as a model for screening. We have not pursued using the other cell lines to test how molecularly they respond to SCH772984.

Also, TNF – which one or the assay is not specific for particular member of TNF family?

Response: We used TNFalpha assay and now refer to the TNFa accordingly through the manuscript

2. Therefore, the authors have used cancer cell line to investigate SCH772984 treatment (with or without LPS) and for further screening and analysis. Data from the airway epithelial cells which represents a primary site for the entry of pathogenic bacteria into the lungs or kidney epithelial cells would be highly informative.

Response: Carrying out the sepsis studies with the other models of bacteria entry as suggested by the Referee would require another major effort and resources and as such should be planned for the future follow up experiments once our data on SCH772984 is released. We discuss the future directions and limitations of our study in the conclusion section.

3. Why is ERK 1/2 phosphorylation decreased in quiescent RAW264.7 upon the treatment with SCH772984 (Fig 1B)? How many samples/per group do you have for Western blot analysis? Could you show densitometry graph?

Response: In the original Fig 1B panel we mistakenly included a result for total ERK. In revised version we include pERK and provide the densitometric analysis from two independent experiments.

4. It is very hard to read text (gene clusters) in the Figure 2B.

Response: The text has been enlarged by 50%

5. What’s happening with TNF in mouse plasma LPS or CLP model after treatment with SCH772984?

Response: We have not assessed the TNFa in the LPS survival experiment. The data for TNFa and the other mediators were assessed in 6h/12h CLP experiment are now provided in Figure S3.

6. Western blot of pERK1/2 in mouse tissue (at least lung and liver tissue) should be shown.

Response: We included the WB data for three liver and lung tissues per trait on Figure S4. There are no significant differences in the pERK1/2 level in those organs following SCH772984 treatment.

7. What is happening with the levels of inflammatory mediators most relevant in sepsis as TNFα, IL-1β, and IL-6 in LPS and CLP mouse model with and without treatment with SCH772984?

Response: See response to point 5

Minor points:

9. Line 41-42, it’s stated: “outcome and determines transcriptome signatures of SCH772984 action in vitro and a preclinical model of sepsis.” I think it will be better to change in: … outcome and determines transcriptome signatures of SCH772984 action in vitro and in vivo, in a preclinical model of sepsis.

Response: Corrected as suggested.

10. Methodology should start with small molecule screening following by in vitro cell work and then in vivo mouse models.

Response: The methods section has been rearranged as suggested by both Referees

Reviewer 2 Report

The authors investigated the effects of the compound, SCH772984, against sepsis using in vitro and in vivo models. The study was well-planned, the results were well organized and presented with figures, and the findings were discussed thoroughly. The manuscript can be improved by the following 8 bullet points so that the readers can follow the study easily.

1) The results were presented in good order, however, difference sub-sections can be created for improved clarity.

My recommendations (the authors should revise the wordings/terms used):

3.1 Cell line models for in-vitro studies

Lines 265-271

3.2 Compounds screening for inhibition of TNF production

Lines 271-292

3.3 The effect of the compound SCH772984 against LPS challenge in cell line RAW 264.7

3.3.1 Phosphorylation of ERK1/2

Lines 292-293

3.3.2 Expression of inflammatory genes

Lines 293-295

3.3.3 Transcriptomes analysis

Lines 305-332

3.3.4 Chromatin accessibility

Lines 333-352

3.4 The effect of the compound SCH772984 in mouse models

3.4.1 Survival experiment

3.4.1.1 LPS-induced sepsis

Lines 356-358

3.4.2.2 CLP-induced sepsis

Lines 359

3.4.2 Immune response in CLP-induced sepsis model

3.4.2.1 Cytokine/chemokines levels in plasma

Lines 360-373

3.4.2.2 Transcriptomes analysis in lung, kidney, liver, and heart

Lines 382-450

2) Similarly, the method sections can be revised accordingly. The methods described should be at the same/similar levels with the results section. Then, the readers will not be lost easily and can correspond the results and methods in the same levels:

2.1 and 3.1: cell line models

2.2 and 3.2: compounds screening

2.3 and 3.3: SCH772984 against LPS challenge in RAW 264.7

2.3.1 and 3.3.1: ERK1/2

2.3.2 and 3.3.2: inflammatory genes

2.3.3 and 3.3.3: transcriptomes

2.3.4 and 3.3.4: chromatin accessibility

2.4 and 3.4: SCH772984 in mouse models

2.5 – methods shared by sections 2.3.2 and 2.3.3

2.6 – methods shared by sections 2.3.3 and 2.4.5.2

My recommendations (the authors should revise the wordings/terms used):

2.1 Evaluation of cell line models for in-vitro studies

Lines 133-136

2.2 Screening of compounds for inhibition of TNF production

Lines 137-154

2.3 Assessing the compound SCH772984 against LPS challenge in cell line RAW 264.7

2.3.1 Protein expression of ERK1/2

Western blot

Lines 168-180

2.3.2 RT-qPCR of inflammatory genes

RNA isolation was described in section 2.5.

(RT-q)PCR

Lines 190-200

2.3.3 Transcriptomes analysis

RNA isolation was described in section 2.5.

Transcriptomes analysis were described in section 2.6.

2.3.4 ATAC-Seq

Lines 220-258

2.4 Assessing the effect of compound SCH772984 in mouse models

2.4.1 Mice

Lines 103-106

2.4.2 LPS-induced sepsis

Lines 108-109

2.4.3 CLP-inducted sepsis

Lines 109-118

2.4.4 Survival experiment

Lines 118-119, 123-127

2.4.5 Investigation of immune response in CLP-induced sepsis model

2.4.5.1 Cytokine/chemokines levels in plasma

Lines 128-129, 161-167

2.4.5.2 Transcriptomes in lung, kidney, liver, and heart

Lines 130-132

Transcriptomes analysis were described in section 2.6.

2.5 RNA isolation

Lines 181-189

2.6 RNA-Seq data analysis and functional analyses of the transcriptomic data

Lines 201-219

3) The authors are recommended to illustrate the study design in one figure, the figure can be either shown in the main text or in supplementary section.

4) Lines 32-33: in in-vitro model, the compound SCH772984 was treated first and then LPS was challenged after. Lines 35-38: In in-vivo model, either LPS-induced sepsis or CLP-inducted sepsis was introduced first and then the drug SCH772984 was administrated later.

Please explain the drug was treated first in in-vitro model while the drug was treated later in in-vivo model.

5) Survival experiment, lines 123-128:

- LPS-induced sepsis model: LPS introduced first, 2 hours later, the drug SCH772984 was administrated, then every 6 hours later the drug SCH772984 was administrated.

- It means that the drug was administrated at the following time points 2h, 8h, 14h, 20h, 26h, 32h, 38h…….until?

- Mortality was recorded at when (Figures 3A and 3B)?

- The same procedures for CLP-induced sepsis model?

6) Table 1 should be revised:

- Protein name > Host protein

- Compound name > Compound targeting the host protein

- Add a new column: Mechanism of actions of the compound (fill in Kinases, Histone deacetylases………….)

- TNF………(add the words ‘in cell line RAW 264.7)

7) The authors can mention the limitations of the current study and field(s) to be investigated in future in the discussion section.

8) Minor issues/typo mistakes

- Line 381: need to describe: ‘only 7 cytokines/chemokines with significant alternations to plasma levels were shown’.

- Line 394: typo mistake, Figure 4B, should be Figure 4A.

- Figure 4B: near the 12 h bar, the word ‘organ’ should be moved to the right hand side to decode the four colors bar. On the other hand, need to add explanations of another bar ’10, 0, -1, -2, -3’. I cannot find any descriptions of this bar. I guess it should be fold change, right? (line 398).

- Figure 4 (CLP vs sham-operated) and Figure 5 (CLP vs CLP drug treated) can be combined together for easy comparison. It means that 4A and 5A should be put side by side, 4B and 5B to put side by side).

- Then, the authors then do not need to describe the footnotes 2 times in Figure 4A and Figure 5A.

- The scales of fold change should be concordant in Figure 4A (10, 0, -1, -2, -3) and Figure 5A (10, 5, 0, -5, 10).

- For the 27 pathways in Figure 4B and 64 pathways in Figure 5B, try to combine, re-organize and list the pathways mentioned in either Figure 4B or Figure 5B. If that pathway was not meet the requirement to show in either Figure 4B or Figure 5B (line 400), use another color to show. Then, readers will observe the differences between ‘CLP vs sham-operated’ and ‘CLP vs CLP drug treated’ easily.

Author Response

Rewiever#2:

The authors investigated the effects of the compound, SCH772984, against sepsis using in vitro and in vivo models. The study was well-planned, the results were well organized and presented with figures, and the findings were discussed thoroughly. The manuscript can be improved by the following 8 bullet points so that the readers can follow the study easily.

1) The results were presented in good order, however, difference sub-sections can be created for improved clarity.

My recommendations (the authors should revise the wordings/terms used):

3.1 Cell line models for in-vitro studies

Lines 265-271

3.2 Compounds screening for inhibition of TNF production

Lines 271-292

3.3 The effect of the compound SCH772984 against LPS challenge in cell line RAW 264.7

3.3.1 Phosphorylation of ERK1/2

Lines 292-293

3.3.2 Expression of inflammatory genes

Lines 293-295

3.3.3 Transcriptomes analysis

Lines 305-332

3.3.4 Chromatin accessibility

Lines 333-352

3.4 The effect of the compound SCH772984 in mouse models

3.4.1 Survival experiment

3.4.1.1 LPS-induced sepsis

Lines 356-358

3.4.2.2 CLP-induced sepsis

Lines 359

3.4.2 Immune response in CLP-induced sepsis model

3.4.2.1 Cytokine/chemokines levels in plasma

Lines 360-373

3.4.2.2 Transcriptomes analysis in lung, kidney, liver, and heart

Lines 382-450

Response: We appreciate the effort the Referee made with this suggestion. However, we believe that the current titles of the subsections better convey the results and we would like to keep them that way. However we agree with some of the suggestions of rearranging the methods section.

2) Similarly, the method sections can be revised accordingly. The methods described should be at the same/similar levels with the results section. Then, the readers will not be lost easily and can correspond the results and methods in the same levels:

2.1 and 3.1: cell line models

2.2 and 3.2: compounds screening

2.3 and 3.3: SCH772984 against LPS challenge in RAW 264.7

2.3.1 and 3.3.1: ERK1/2

2.3.2 and 3.3.2: inflammatory genes

2.3.3 and 3.3.3: transcriptomes

2.3.4 and 3.3.4: chromatin accessibility

2.4 and 3.4: SCH772984 in mouse models

2.5 – methods shared by sections 2.3.2 and 2.3.3

2.6 – methods shared by sections 2.3.3 and 2.4.5.2

My recommendations (the authors should revise the wordings/terms used):

2.1 Evaluation of cell line models for in-vitro studies

Lines 133-136

2.2 Screening of compounds for inhibition of TNF production

Lines 137-154

2.3 Assessing the compound SCH772984 against LPS challenge in cell line RAW 264.7

2.3.1 Protein expression of ERK1/2

Western blot

Lines 168-180

2.3.2 RT-qPCR of inflammatory genes

RNA isolation was described in section 2.5.

(RT-q)PCR

Lines 190-200

2.3.3 Transcriptomes analysis

RNA isolation was described in section 2.5.

Transcriptomes analysis were described in section 2.6.

2.3.4 ATAC-Seq

Lines 220-258

2.4 Assessing the effect of compound SCH772984 in mouse models

2.4.1 Mice

Lines 103-106

2.4.2 LPS-induced sepsis

Lines 108-109

2.4.3 CLP-inducted sepsis

Lines 109-118

2.4.4 Survival experiment

Lines 118-119, 123-127

2.4.5 Investigation of immune response in CLP-induced sepsis model

2.4.5.1 Cytokine/chemokines levels in plasma

Lines 128-129, 161-167

2.4.5.2 Transcriptomes in lung, kidney, liver, and heart

Lines 130-132

Transcriptomes analysis were described in section 2.6.

2.5 RNA isolation

Lines 181-189

2.6 RNA-Seq data analysis and functional analyses of the transcriptomic data

Lines 201-219

3) The authors are recommended to illustrate the study design in one figure, the figure can be either shown in the main text or in supplementary section.

Response: we prepared a graphical abstract that summarizes the study

4) Lines 32-33: in in-vitro model, the compound SCH772984 was treated first and then LPS was challenged after. Lines 35-38: In in-vivo model, either LPS-induced sepsis or CLP-inducted sepsis was introduced first and then the drug SCH772984 was administrated later.

Please explain the drug was treated first in in-vitro model while the drug was treated later in in-vivo model.

Response: The preventive or therapeutic treatment are two options of drugs administration. For in vitro we used the preventive approach as it is more feasible to synchronize cells in a high throughput experiment. On the other hand, the therapeutic intervention, following the induction of sepsis is more relevant to the day-to-day situation when therapeutic decisions are made based on clinical presentation.

5) Survival experiment, lines 123-128:

- LPS-induced sepsis model: LPS introduced first, 2 hours later, the drug SCH772984 was administrated, then every 6 hours later the drug SCH772984 was administrated.

  • It means that the drug was administrated at the following time points 2h, 8h, 14h, 20h, 26h, 32h, 38h…….until?

Response: We includes the description in LPS survival experiment “then every 6 hours later the drug SCH772984 was administrated “

- Mortality was recorded at when (Figures 3A and 3B)?

- The same procedures for CLP-induced sepsis model?

Response: We included sepsis severity scoring system description used for the euthanasia criterion.

6) Table 1 should be revised:

- Protein name > Host protein

- Compound name > Compound targeting the host protein

- Add a new column: Mechanism of actions of the compound (fill in Kinases, Histone deacetylases………….)

  • TNF………(add the words ‘in cell line RAW 264.7)

Response: Corrected

7) The authors can mention the limitations of the current study and field(s) to be investigated in future in the discussion section.

Response: We discuss the future directions and limitations of our study in the conclusion section

8) Minor issues/typo mistakes

- Line 381: need to describe: ‘only 7 cytokines/chemokines with significant alternations to plasma levels were shown’.

Response: Corrected

- Line 394: typo mistake, Figure 4B, should be Figure 4A.

- Figure 4B: near the 12 h bar, the word ‘organ’ should be moved to the right hand side to decode the four colors bar. On the other hand, need to add explanations of another bar ’10, 0, -1, -2, -3’. I cannot find any descriptions of this bar. I guess it should be fold change, right? (line 398).

- Figure 4 (CLP vs sham-operated) and Figure 5 (CLP vs CLP drug treated) can be combined together for easy comparison. It means that 4A and 5A should be put side by side, 4B and 5B to put side by side).

- Then, the authors then do not need to describe the footnotes 2 times in Figure 4A and Figure 5A.

- The scales of fold change should be concordant in Figure 4A (10, 0, -1, -2, -3) and Figure 5A (10, 5, 0, -5, 10).

- For the 27 pathways in Figure 4B and 64 pathways in Figure 5B, try to combine, re-organize and list the pathways mentioned in either Figure 4B or Figure 5B. If that pathway was not meet the requirement to show in either Figure 4B or Figure 5B (line 400), use another color to show. Then, readers will observe the differences between ‘CLP vs sham-operated’ and ‘CLP vs CLP drug treated’ easily.

Response: thank you for this suggestion. Combining the figures allowed us to capture additional, important pathways, namely the ECM and retinoic acid pathways that are positively regulated by the SCH772984.

Round 2

Reviewer 1 Report

The authors have answered my concerns and improved the manuscript considerably. Yet, few minor comments remain:

  1. Figure 1B – As this Western blot is important, at least 3 experiments should be done to include statistical analysis for densitometry, which cannot be done with 2 samples/group.
  2. Figure 2B – I appreciate that letter size was increased in the Figure, but that’s not necessarily resulted in improved Figure clarity. I recommend to put some abbreviations that could be then explained in the Figure Legend (i.e. Antigen should be abbreviated as Ag; extracellular as E.C.; extracellular matrix as E.C. M.), Interleukin as IL, etc.
  3. Figure 3C, it’s very hard to read the group name or unit; letter size should be increased.

Author Response

  1. Figure 1B – As this Western blot is important, at least 3 experiments should be done to include statistical analysis for densitometry, which cannot be done with 2 samples/group.

Response: In revised version we provide data from four experiments. The statistical analysis of densitometric data showed significant inhibition of pERK abundance at 1h and 4h LPS time point upon SCH772984 treatment.

  1. Figure 2B – I appreciate that letter size was increased in the Figure, but that’s not necessarily resulted in improved Figure clarity. I recommend to put some abbreviations that could be then explained in the Figure Legend (i.e. Antigen should be abbreviated as Ag; extracellular as E.C.; extracellular matrix as E.C. M.), Interleukin as IL, etc.

Response: We have incorporated suggested abbreviations to improve the figure clarity.

  1. Figure 3C, it’s very hard to read the group name or unit; letter size should be increased.

Response: We have enlarged by 50% protein names, unit and groups names.

Reviewer 2 Report

The authors addressed all of my queries.

Author Response

There were no further comments from this Referee.

Thank you!